# Growth of *Listeria monocytogenes* in Partially Cooked Battered Chicken Nuggets as a Function of Storage Temperature

**DOI:** 10.3390/foods10030533

**Published:** 2021-03-04

**Authors:** Alexandra Lianou, Ourania Raftopoulou, Evgenia Spyrelli, George-John E. Nychas

**Affiliations:** 1Laboratory of Microbiology and Biotechnology of Foods, Department of Food Science and Human Nutrition, School of Food and Nutritional Sciences, Agricultural University of Athens, 11855 Athens, Greece; raftopoulourania@gmail.com (O.R.); eugeniespcheng@gmail.com (E.S.); 2Division of Genetics, Cell Biology and Development, Department of Biology, University of Patras, 26504 Patras, Greece; 3Department of Food, Bioprocessing and Nutrition Sciences, North Carolina State University, Raleigh, NC 27695-7624, USA

**Keywords:** *Listeria monocytogenes*, battered chicken nuggets, growth, modelling, temperature

## Abstract

Battered poultry products may be wrongly regarded and treated by consumers as ready-to-eat and, as such, be implicated in foodborne disease outbreaks. This study aimed at the quantitative description of the growth behavior of *Listeria monocytogenes* in fresh, partially cooked (non-ready-to-eat) battered chicken nuggets as function of temperature. Commercially prepared chicken breast nuggets were inoculated with *L. monocytogenes* and stored at different isothermal conditions (4, 8, 12, and 16 °C). The pathogen’s growth behavior was characterized via a two-step predictive modelling approach: estimation of growth kinetic parameters using a primary model, and description of the effect of temperature on the estimated maximum specific growth rate (*μ_max_*) using a secondary model. Model evaluation was undertaken using independent growth data under both constant and dynamic temperature conditions. According to the findings of this study, *L. monocytogenes* may proliferate in battered chicken nuggets in the course of their shelf life to levels potentially hazardous for susceptible population groups, even under well-controlled refrigerated storage conditions. Model evaluation demonstrated a satisfactory performance, where the estimated bias factor (*B_f_*) was 0.92 and 1.08 under constant and dynamic temperature conditions, respectively, while the accuracy factor (*A_f_*) value was 1.08, in both cases. The collected data should be useful in model development and quantitative microbiological risk assessment in battered poultry products.

## 1. Introduction

*Listeria monocytogenes* is a foodborne pathogen of ongoing public health significance [1]. Despite its relative rare occurrence, human listeriosis is among the most serious foodborne diseases under European Union (EU) surveillance, being associated with high morbidity and mortality, particularly among the elderly. A notification rate of 0.47 cases per 100,000 population and a case fatality rate as high as 15.6% were reported in 2018 in the EU [1].

Due to its ubiquitous presence in the natural environment and its psychrotrophic character, *L. monocytogenes* is an important hazard as post-processing contaminant for refrigerated food products. Given the well-established positive association of processed meat and poultry products with the risk of foodborne listeriosis [2,3], assessing the incidence and fate of *L. monocytogenes* in various ready-to-eat (RTE) products of this category has been the objective of a large volume of research studies globally [4,5,6,7,8]. With specific reference to poultry products, the growth behavior of this pathogenic bacterium has been evaluated in different RTE products such as cooked chicken meat, turkey breast and turkey bologna [9,10,11]. On the other hand, assessing the pathogen’s fate in raw or partially cooked poultry products has not been a research priority, mainly due to the so far low association of such products (which are expected to be sufficiently cooked prior to consumption) with foodborne listeriosis [12]. Nonetheless, as indicated by quantitative microbiological risk assessment (QMRA) findings, raw meat and poultry products can also be the cause of listeriosis if they are not stored, cooked, and/or handled properly [13], with consumer behavior being identified as playing a central role in QMRA in poultry meat [12].

Both sporadic and epidemic listeriosis have been associated with poultry production and processing environments as well as with various poultry products [14,15]. Beyond live birds, which constitute an important reservoir of *L. monocytogenes* [15], reports on high prevalence of the pathogen in raw chicken products, along with concerns of increasing antibiotic resistance [4,16,17,18,19], demonstrate that its association with poultry meat production should not be regarded as trivial from a public health perspective. Indeed, *L. monocytogenes* may establish niches in poultry slaughterhouses and processing environments, serving as persistent sources of cross-contamination of end products, especially under conditions of poor compartmentalization of the processing line [18,20,21]. Non-RTE chicken nuggets and other breaded chicken products have actually been recalled due to possible contamination with the pathogen [22,23]. Although battering and breading may enhance microbial heat resistance [24], sufficient cooking should ensure elimination of *L. monocytogenes* in poultry products [25,26]. Yet, as suggested by epidemiological investigations, cooked poultry products may also be involved in listeriosis outbreaks [27], whereas consumers may wrongly regard and treat battered/breaded chicken products as fully cooked (due to the applied surface browning step) and only reheat them before consumption [28,29,30,31,32], thus not ensuring elimination of pathogenic bacteria [33]. In the event of such consumer malpractices, the presence of *L. monocytogenes* in chicken nuggets, either as a raw-ingredient contaminant surviving the manufacturing process or as a post-processing contaminant, may compromise their safety, especially under abusive temperature conditions which are rather common along the food cold chain [34,35].

Although the survival, growth, and control of *Salmonella enterica* in non-RTE poultry-based meat preparations, including breaded chicken nuggets, have been investigated [32,33,36,37], pertinent research data for *L. monocytogenes* are scarce. Certainly, *L. monocytogenes* growth has been mainly evaluated in delicatessen and other RTE poultry products [9,10,11], with only a few studies in the scientific literature reporting on its fate in poultry-based nugget products [38,39]. Yet, the growth behavior of *L. monocytogenes* in partially cooked (i.e., non-RTE) chicken nuggets has not been previously assessed.

Given the above, the objective of the present study was the assessment and quantitative description of the growth behavior of *L. monocytogenes* in partially cooked battered chicken nuggets as a function of storage temperature.

## 2. Materials and Methods

### 2.1. Listeria monocytogenes Strains and Inoculum Preparation

The *L. monocytogenes* strains used in the present work were the following: B-124 (designation NCTC 10527, clinical isolate, serotype 4b), B-125 (designation Scott A, clinical isolate, serotype 4b), B-127 (designation 21075, chicken salad isolate), B-129 (designation 21350, RTE frozen, meat-based meal isolate), B-131 (designation 214112, conveyor belt isolate), and B-157 (designation 23UD, food processing plant isolate). Strain B-125 was originally provided by Dr. Eddy Smid (Agrotechnological Research Institute ATO-DLO, Wageningen, the Netherlands) [40], while strain B-157 by Dr. Luca Cocolin (University of Turin, Turin, Italy) [41]. The abovementioned strains were used as a multiple-strain composite in this study in order to account for variation in growth among *L. monocytogenes* strains, in line with pertinent recommendations and guidelines with regard to food safety challenge studies [42,43,44].

The maintenance (frozen stock and working cultures) and revival of the *L. monocytogenes* strains, as well as the procedures followed for inoculum preparation, were the same as the ones reported previously [45]. In brief, for the purpose of inoculum preparation, the *L. monocytogenes* strains were activated (30 °C for 24 h) and sub-cultured (30 °C for 18 h) in tryptic soy broth (Biolife, Milan, Italy), combined in two three-strain mixtures and the latter were centrifuged (Heraus Multifuge 1S–R, Thermo Electron Corporation, Langenselbold, Germany) at 6000 rpm for 20 min at 4 °C. The harvested cells were washed with quarter strength Ringer’s solution (Lab M Limited, Lancashire, UK) via centrifugation under the same conditions, and were resuspended in 30 mL of Ringer’s solution, and the strains’ suspensions were mixed. The six-strain mixture was diluted in Ringer’s solution to yield an inoculum concentration of 1–2 log CFU/g of chicken nuggets when a total of 300 μL of inoculum was applied to a nugget. For the purpose of determining its exact initial concentration and also checking its purity, appropriate decimal dilutions in Ringer’s solution of the prepared inoculum were surface plated on polymyxin-acriflavine-LiCl-ceftazidime-aesculin-mannitol (PALCAM) agar (basal medium and selective supplement, Biolife) and tryptic soy agar (TSA, Biolife).

### 2.2. Inoculation of Chicken Nuggets and Storage Conditions

The product studied in the present work was fresh, battered, and partially cooked (i.e., pre-cooked but non-RTE) chicken breast nuggets, provided by a local poultry processor. The product’s ingredients include chicken breast (80%) marinated in a mixture of herbs and spices, and batter (20%) being composed of water, wheat starch, corn starch, modified wheat starch, salt, soy protein, soybean oil, egg powder, and bulking agents (sodium carbonate and disodium diphosphate). The product is commercially packaged in plastic trays heat-sealed with a plastic film, its shelf life is 10 days (from the day of production) under refrigeration (0–4 °C), and according to its preparation instructions, it should be baked or fried at 180 °C for 10 or 6 min, respectively, prior to consumption. The chicken nuggets were provided by the processor and inoculated in the laboratory within 24 h from manufacture.

Individual nuggets weighing ca. 30 g (ranging from 28.5 to 31.5 g) were inoculated with *L. monocytogenes* inside a biological safety cabinet. Specifically, 150-μL aliquots of the six-strain mixture of the pathogen were applied via spot inoculation (using a pipette) throughout the surface of one side of each chicken nugget, the nuggets were maintained at 4 °C for 10 min for inoculum attachment, and then the same procedure was repeated for the inoculation of the other side. Afterwards, chicken nuggets were aseptically transferred to Styrofoam trays (two nuggets per tray), which were wrapped with cling film (i.e., polyethylene and oxygen-permeable household food wrap) and stored at different isothermal conditions (4, 8, 12, and 16 °C) in high-precision (±0.5 °C) programmable incubators (MIR-153, Sanyo Electric Co., Osaka, Japan). The incubation temperatures were recorded at 20-min intervals throughout storage using electronic temperature-monitoring devices (COX TRACER^®^, Cox Technologies Inc., Belmont, NC, USA).

Battered chicken nuggets corresponding to two distinct product batches were artificially inoculated and stored as described above, in the course of two independent experimental replicates conducted at different time periods, allowing for biological variation to be taken into account. Uninoculated product samples were also used in the framework of the applied experimental conditions for the purpose of: (i) assessing the natural occurrence (at detectable levels) of *L. monocytogenes* and (ii) providing complementary information on the natural microbiota of the product.

### 2.3. Microbiological Analyses

Microbiological analyses of chicken nuggets were performed directly after inoculations (time-zero), as well as during storage at time intervals depending on the applied temperature. For this purpose, samples (individual intact chicken nuggets) were transferred aseptically to sterile stomacher bags (Baglight^®^, Interscience, Saint-Nom-la-Bretèche, France), were 1:10 diluted in sterilized Ringer’s solution, were homogenized in a Stomacher apparatus (Lab Blender 400, Seward Medical, London, UK) for 60 s, and appropriate serial decimal dilutions in Ringer’s solution were surface plated on the following agar media: (i) PALCAM agar (basal medium and selective supplement) for the determination of *L. monocytogenes* populations after incubation of plates at 30 °C for 48 h; (ii) TSA for the determination of total mesophiles’ populations after incubation of plates at 30 °C for 72 h; and (iii) Rose Bengal Chloramphenicol (RBC) agar (Lab M Limited) for the determination of the populations of molds and yeasts after incubation of plates at 25 °C for 5 days (at selected time intervals during storage). The microbiological data obtained after enumeration of the microbial colonies grown on the agar media were converted to log CFU/g.

With regard to the uninoculated chicken nuggets, which were stored in parallel under the same conditions (and corresponded to the same product batches as the artificially inoculated samples), these were analyzed according to the aforementioned procedures for the determination of the populations of total mesophiles (on TSA), molds and yeasts (on RBC agar), and for the natural occurrence (at detectable levels), if any, of *L. monocytogenes* (on PALCAM agar). Furthermore, the uninoculated product samples were analyzed for the incidence (if present at detectable levels) of *Salmonella* spp. via surface plating on xylose lysine deoxycholate (XLD) agar (Neogen^®^ Culture Media, Lansing, MI, USA), as well as for the presence/enumeration of bacteria of the Enterobacteriaceae family by pour plating in violet red bile glucose (VRBG) agar (Oxoid Ltd., Basingstoke, Hampshire, UK). The XLD and VRBG agar plates were examined for bacterial growth after incubation at 37 °C for 24 h.

During storage and upon completion of the microbiological analyses, the pH values of the chicken nuggets also were measured using a digital pH meter (RL150, Russell pH, Cork, Ireland) with a glass electrode (Metrohm AG, Herisau, Switzerland).

Within each one of the two independent experimental replicates (distinct product batches), duplicate chicken nuggets were analyzed at each sampling time (*n* = 4) during storage at the different temperature conditions. In this framework, a total of 264 inoculated (with *L. monocytogenes*) and 172 uninoculated battered chicken nuggets were analyzed during isothermal storage at different temperatures.

### 2.4. Modelling of L. monocytogenes Growth

Aiming at the quantitative description of *L. monocytogenes* growth in battered chicken nuggets, a two-step predictive modelling approach was applied as previously described [45]. Specifically, the pathogen’s growth kinetic behavior was characterized via the primary model of Baranyi and Roberts [46], which was fitted to the collected microbiological data (the natural logarithm of the pathogen’s counts) using the Microsoft^®^ Excel Add-in curve-fitting program DMFit, Version 3.5 (Institute of Food Research, Norwich, UK). The pathogen’s growth kinetic parameters, namely the lag time (*λ*), the maximum specific growth rate (*μ_max_*), and the maximum population density (*y_end_*) corresponding to the upper asymptote of the sigmoid curve, were estimated for each one of the tested product samples and at each one of the applied storage temperatures. The effect of temperature on *μ_max_* was then modeled using the following square-root-type model [47]:(1)$$\sqrt{\mu_{max}}=b\left(T-T_{min}\right)$$
where *b* is a constant, *T* is the temperature (°C), and *T_min_* is a theoretical minimum temperature for microbial growth (i.e., intercept between the model and the temperature axis). The values of *b* and *T_min_* were determined by fitting the above model to the estimated *μ_max_* values with linear regression using Microsoft^®^ Excel (Microsoft Corp., Redmond, WA, USA).

Model evaluation (i.e., external validation) was based on independent growth data, generated during storage of inoculated battered chicken nuggets under both isothermal and dynamic temperature conditions. Specifically, additional chicken nuggets (corresponding to different product batches) were artificially inoculated with *L. monocytogenes* according to the procedures described in Section 2.2, and then stored at: (i) the constant temperature of 10 °C (i.e., within the range of the temperatures covered by the model but not actually used in its calibration), and (ii) non-isothermal temperature conditions delineated by periodic temperature changes from 4 to 12 °C (8 h at 4 °C, 8 h at 8 °C, and 8 h at 12 °C). The temperatures encountered during the applied isothermal and dynamic temperature storage experiments were also recorded using electronic data loggers, as described above. Under both storage temperature scenarios, duplicate chicken nuggets were analyzed at regular time intervals for the determination of *L. monocytogenes* populations, as described in Section 2.3. Overall, for model validation purposes, 60 (two independent experimental replicates corresponding to two distinct product batches) and 30 (one experimental replicate) product samples were analyzed during storage at 10 °C and dynamic temperatures, respectively. The pathogen’s growth was predicted using: (i) the acquired time–temperature profiles; (ii) the above mentioned secondary model for the estimation of the *μ_max_* at 10 °C or the “momentary” *μ_max_* in the case of the dynamic temperature scenario; and (iii) the differential equations of the Baranyi and Roberts model [46,48], which were numerically integrated using Microsoft^®^ Excel, as described in detail previously [45,49]. Model performance was assessed graphically, as well as numerically using the performance indices of bias factor (*B_f_*) and accuracy factor (*A_f_*) [50], based on the predicted vs. observed generation times (*GT* = ln2/*μ_max_*). In the case of the dynamic temperature conditions, the *μ_max_* values used in the determination of the aforementioned performance metrics were estimated via fitting of the predicted and observed growth data to the primary model of Baranyi and Roberts, assuming a constant value for this growth parameter.

## 3. Results and Discussion

### 3.1. Microbial Growth Data

The *L. monocytogenes* populations in battered chicken nuggets (as enumerated on PALCAM agar) under the different tested isothermal conditions are shown in Figure 1. The mean (±standard deviation (sd), *n* = 4) inoculation level (i.e., time-zero concentration) of the organism attained in the product was 1.15 (±0.17) log CFU/g. As demonstrated both graphically and by the estimated growth kinetic parameters (Table 1), the pathogen’s growth behavior in battered chicken nuggets was, as expected, temperature-dependent with increasing storage temperature resulting in faster growth. Indeed, the mean (±sd, *n* = 4) *L. monocytogenes* population (log CFU/g) recorded at 240 h of storage (corresponding to the end of the product’s commercial shelf life) was 4.04 (±0.40), 7.05 (±1.65), 8.90 (±0.64), and 9.04 (±0.67) at 4, 8, 12, and 16 °C, respectively (Figure 1). The significant effect of temperature on *μ_max_* was also demonstrated by the results of one-way analysis of variance, according to which the estimated value of the test statistic was F (3,12) = 12.49 (*p* = 0.001).

The microbiological data collected under the conditions of this study (and for the tested *L. monocytogenes* strains) demonstrate that, even in the case of relatively low-level contamination and under well-controlled refrigeration (simulated by the isothermal storage at 4 °C herein), *L. monocytogenes* may proliferate in battered chicken nuggets in the course of their shelf life to levels potentially hazardous for susceptible population groups, particularly in the case of highly virulent strains of the organism [14,51]. Moreover, in the case of temperature abuse which is commonly encountered in the food cold chain [34,35], the pathogen may grow to levels capable of causing foodborne listeriosis even among healthy individuals, as illustrated by its growth behavior recorded at the abusive temperature of 8 °C (or higher) in the present study (Figure 1). Since battered/breaded chicken nuggets are often partially cooked (i.e., surface browned) and intended to be fully cooked prior to consumption, the application of an adequate heat treatment (either through baking or frying) is anticipated to ensure the product’s microbiological safety [25]. Nonetheless, consumers may not follow label instructions for cooking treatments and engage to unsafe practices such as undercooking [12] or no cooking at all considering such products as RTE.

With reference to the naturally occurring microbiota of the studied poultry product, this appeared to be comprised predominantly by psychrotrophic yeasts. In the artificially inoculated chicken nuggets, the TSA microbial counts were higher than the PALCAM agar counts, with the recorded difference being reduced as storage temperature increased (Figure 1). Furthermore, the TSA counts obtained at all storage temperatures were similar to the microbial counts determined on RBC agar with the latter corresponding solely to colonies with typical yeast-like appearance (i.e., no mycelial growth was noted). Actually, the yeasts’ growth during storage of the chicken nuggets at refrigeration (4 °C) and slightly abusive temperatures (8 °C), as recorded in the present study (and reflected in the total mesophilic microbial populations), was markedly superior to the observed *L. monocytogenes* growth (Figure 1). The predominance, if not exclusive presence, of yeasts in the natural microbiota of the studied chicken nuggets, is also demonstrated by the microbiological analyses’ results of the uninoculated (control) product samples (Figure 2). The yeasts and total mesophiles exhibited similar growth profiles and population levels during storage at all tested temperatures, while the slightly higher microbial counts obtained on TSA (as compared to those on RBC) are, most likely, attributed to the absence of selective agents (potentially acting as stress factors) in this generic medium. The use of RBC agar (over other selective growth media) for the characterization of the chicken nuggets’ natural microbiota was based on the findings of preliminary experiments according to which, yeasts proliferated during storage and constituted the dominant naturally occurring microbial group throughout the product’s shelf life. No detectable populations of naturally occurring *Listeria* spp. were enumerated during storage of the control chicken nuggets throughout storage at none of the tested isothermal conditions, while bacteria belonging to the Enterobacteriaceae family, including specifically *Salmonella* spp., were also below the detection limit (<1 log CFU/g) at all temperatures and sampling intervals. Since psychrophilic bacterial populations were not enumerated in the framework of the present work, future studies, specifically designed and conducted for this purpose, should contribute to the in-depth characterization of the microbial association of this product and its evolution during storage.

The overall microbiological quality of battered chicken nuggets is anticipated to reflect the microbiological status of the raw poultry meat and the rest of the product’s ingredients, as affected by the applied manufacturing processes and prevailing environmental conditions. Taking into account that the chicken nuggets studied herein are a partially cooked product manufactured from marinated chicken breast meat, it can be justified why spoilage bacteria traditionally associated with aerobically stored poultry, such as *Pseudomonas* spp. [19,38,52], did not appear to comprise an eminent part of this product’s natural microbiota. In the same context, the psychrotrophic yeasts, identified under the conditions of this study as the dominant natural microbiota of the studied product, are most likely the result of post-process contamination with species/strains prevalent in the poultry processing environment. Indeed, it has been shown that yeasts may play an important role in the spoilage of fresh and processed poultry during refrigerated storage [53], whereas in the case of battered products, yeast species in the manufacturing environment (and ultimately in the final product) may also originate from the used batter and its ingredients [54]. Finally, the practical absence of background competing bacterial microbiota in battered chicken nuggets, as demonstrated under the conditions of this study, may also be regarded as a risk factor for the growth of *L. monocytogenes* to potentially hazardous levels over the product’s shelf life. Certainly, the evolution of the autochthonous microbiota of meat products, regulated among others by the applied temperature and packaging conditions, may in turn considerably affect (either positively or negatively) the organism’s growth potential [38,55].

Regarding the pH of battered chicken nuggets, the mean (±sd, *n* = 4) value of the inoculated with *L. monocytogenes* samples was 6.21 (±0.36) on day-0, it was not considerably changed during storage at 4 °C, and it was reduced to 5.82 (±0.09), 5.82 (±0.05), and 5.88 (±0.12) during storage at 8, 12, and 16 °C, respectively. A similar situation concerning the initial pH value and its evolution during storage was recorded for the uninoculated product samples.

### 3.2. Modelling of L. monocytogenes Growth Rate

Although the *μ_max_* parameter was estimated via primary modelling at all studied temperatures (i.e., 4, 8, 12, and 16 °C) and for all sample and experimental replicates, *λ* could only be determined for storage at 4 and 8 °C (Table 1), due to the rather fast growth of *L. monocytogenes* at the higher storage temperatures. With reference to the goodness-of-fit of the applied secondary model, the coefficient of determination (*R*^2^), the root mean square error (RMSE) and the standard error (of fit) was 0.920, 0.0214, and 0.0302, respectively. Concerning the values of the model parameters, the constant parameter *b* was 0.0162 (standard error: 0.0003), whereas *T_min_* was estimated to be −9.1 °C (standard error: ± 4.1 °C). The estimated value of the secondary model’s parameter *T_min_* was, certainly, rather low. However, when evaluating the biological significance of such estimate, it should be kept in mind what this parameter value represents. Since *T_min_* is graphically defined as the intercept between the secondary model and the temperature axis, its estimated value is only a theoretical one and may be unrealistically low from a biological standpoint. Actually, it has been presumed that the value of this parameter can be 5 to 10 °C lower than the minimum temperature at which microbial growth is actually observed [56]. Beyond this presumption, the *T_min_* value estimated herein should be appraised taking also into account the low inoculation level utilized in this study (which could have been even below 10 cells/g in some instances). Such relatively low inoculation levels, albeit very realistic, may be rather challenging from a technical and analysis/interpretation perspective, due to the inevitable introduction into the modelling approach of both biological variability and uncertainty. Indeed, it has been shown that bacterial growth behavior tends to be stochastic at low cell concentrations (<100 cells), and the variability in individual cell state and growth kinetic parameters, as well as the uncertainty related to the applied experimental procedures, are of outmost importance in such cases [57,58,59,60]. Nonetheless, artificial inoculation delivering a relatively low initial concentration of the pathogenic bacteria (ca. 10 CFU/g) as practiced in the present study, allows for a better approximation of real-life contamination events than the vast majority of quantitative microbiology studies using initial microbial concentrations in the range of 100–1000 CFU.

For model validation purposes, *L. monocytogenes* growth was predicted under both constant (i.e., 10 °C) and dynamic (periodic changes from 4 to 12 °C) temperature conditions, and it was evaluated against independent growth data acquired at these conditions in the framework of additional storage experiments (i.e., external model validation as described in Section 2.4). It is interesting to note that the rather high *L. monocytogenes* population (ca. 9 log CFU/g) observed at 240 h of storage (approximating the end of the product’s commercial shelf life) at the abusive constant temperatures of 12 °C (Figure 1) and 10 °C (Figure 3), was also attained during storage for the same time period under dynamic temperature conditions (Figure 4). The latter observation is indicative of the detrimental impact that cold chain temperature fluctuations may have on the growth of foodborne pathogens and ultimately, on the safety of chilled food products. Prediction of *L. monocytogenes* growth at isothermal conditions (Figure 3) was performed using the mean temperature recorded during storage, namely 10.65 °C (with the corresponding standard deviation being low and equal to 0.31 °C), the square-root-type model for the determination of the *μ_max_* at this temperature (i.e., 0.102 h^−1^), and the differential equations of the Baranyi and Roberts model which were numerically integrated with respect to time. With regard to growth prediction at dynamic temperature conditions (Figure 4), the same general approach was embraced using: (i) the time–temperature profiles recorded during storage of the samples and (ii) the “momentary” *μ_max_* as determined by the aforementioned secondary model. In both cases, as initial microbial population, was used the mean value of the actual initial inoculum concentrations, determined by plate counting across all inoculation experimental trials including the independent model validation trials (i.e., 1.40 log CFU/g), whereas as maximum population density was used the mean maximum population estimated from the individual curve fittings corresponding to the constant storage temperatures (mean *y_end_* = 8.84 log CFU/g). The “physiological state” parameter *h_0_* was estimated as the product *μ_max_* × *λ* [46], using the values of the growth kinetic parameters determined under the studied isothermal conditions, and specifically the 4 and 8 °C storage temperatures. The *h*_0_ parameter was not considerably different between these two temperatures, with its mean (±sd) value being determined as 1.20 (±0.36) and 1.44 (±0.15) at 4 and 8 °C, respectively. Thus, for the purpose of *L. monocytogenes* growth prediction with the ultimate goal of model validation (both under isothermal and dynamic temperature conditions), *h*_0_ was averaged for temperature and the parameter value used in modelling was *h*_0_ = 1.28.

External model validation against independent isothermal growth data resulted in predictions of *L. monocytogenes* populations close to the actually observed ones. Nonetheless, a systematic under-prediction of the pathogen’s populations was observed (Figure 3). The estimated values of the performance indices of *B_f_* and *A_f_*, were 0.92 and 1.08, respectively. With respect to model evaluation under dynamic temperature conditions, an overall satisfactory performance was attained, with no systematic bias (under- or over-prediction) of the model being evident (Figure 4). In this case, the estimated *B_f_* and *A_f_* was equally 1.08. It has been suggested that *B_f_* values in the range of 0.90–1.05 can be regarded as good for models involving microbial pathogens [61]. Hence, it can be overall claimed that the models’ performance is satisfactory, at least in terms of growth rate, with their tendency, however, to provide fail-dangerous predictions being undeniably a limitation and demonstrating the need for further improvements at both calibration and validation levels.

In both model validation cases (isothermal and dynamic temperature conditions), it can be noted that the fail-dangerous predictions are essentially related to the initial storage period of chicken nuggets, approximately up to 30 h (Figure 3 and Figure 4). This is particularly evident in the case of the isothermal prediction where, as graphically illustrated in Figure 3, the linear part of the predicted growth curve is parallel (i.e., similar slope) to the one of the actually observed growth. The relative incompetence of the developed models to provide accurate predictions at low *L. monocytogenes* population levels (ca. up to 2.5 log CFU/g) may be associated with the low inoculation level utilized in this study and the variability and uncertainty associated with it, as discussed above. Still, and despite the limitations, it is very important that realistically low inoculation levels of foodborne pathogenic bacteria are considered in quantitative microbiology studies. To this end, the present study is among the very few ones providing growth characterization based on low initial cell concentrations. Yet, future research should address challenges encountered when low cell concentrations are studied, through model optimization and fine-tuning attempts based on: (i) better characterization of the biological variability in growth kinetic parameters, particularly in the lag phase of the growth curve, and integration of this variability via stochastic modelling approaches; and (ii) reduction of the uncertainty associated with the applied experimental procedures through the conductance of multiple experimental replicates and the acquisition of larger initial microbial concentration datasets. Moreover, it should be acknowledged that although allowing for the integration of potential growth variation among *L. monocytogenes* strains, the multiple-strain composite utilized in the present study results in model predictions that cannot be linked to specific strain(s) of the organism. Molecular monitoring of the evolution of *L. monocytogenes* strains comprising the applied inoculum in food safety challenge studies has demonstrated that the pathogen’s strains dominating during storage may be considerably affected by the environmental conditions (e.g., temperature), highlighting the importance of strain selection in quantitative microbiology studies [45].

The available growth models involving specifically *L. monocytogenes* and poultry products are relatively limited [11,62,63,64,65,66,67], with both the studied products (mainly raw poultry) and environmental parameters being distinct to the ones evaluated herein. To the best of our knowledge, this is the first study reporting on the assessment and quantitative description of *L. monocytogenes* growth in fresh battered chicken nuggets, a rather popular precooked chicken product. With the majority of the battered/breaded chicken nugget products marketed in the EU and the United States being distributed either as ready-to-cook frozen products or as RTE refrigerated products, the product category investigated herein (i.e., partially cooked but non-RTE product distributed under refrigeration) has not been amply represented in food safety challenge studies. Characterization of *L. monocytogenes* growth as a function of storage temperature, such as the one carried out in this study, should be useful in the development and/or validation of predictive models, with the latter serving as valuable exposure assessment tools. Nevertheless, in order for reliable exposure assessment information to be compiled, the collected data should be evaluated in tandem with data on the thermal inactivation behavior of the pathogen in the studied product. Based on the growth data collected herein, future research should evaluate the thermal inactivation kinetics of *L. monocytogenes* in battered chicken nuggets, considering different scenarios in terms of initial contamination levels and cooking regimes (i.e., time–temperature combinations corresponding both to cooking instructions and undercooking).

## 4. Conclusions

According to the findings of this study, fresh chicken nuggets support the proliferation of *L. monocytogenes*, with the pathogen being capable of reaching potentially hazardous concentration levels in the course of the product’s shelf life, even under well-controlled refrigerated storage conditions. Although further model optimization may be needed, the collected data should contribute to the characterization of *L. monocytogenes* growth in battered chicken nuggets and as such, be useful in quantitative microbiological risk assessment regarding this and similar poultry products.

## Figures and Tables

**Figure 1 foods-10-00533-f001:**
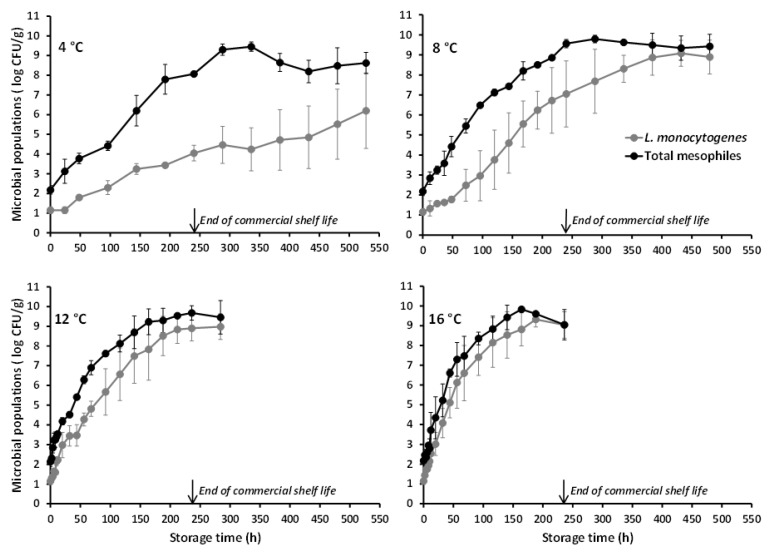
Mean (±standard deviation, *n* = 4) populations of *Listeria monocytogenes* and total mesophiles in battered chicken nuggets, artificially inoculated with the pathogen and stored at different temperatures.

**Figure 2 foods-10-00533-f002:**
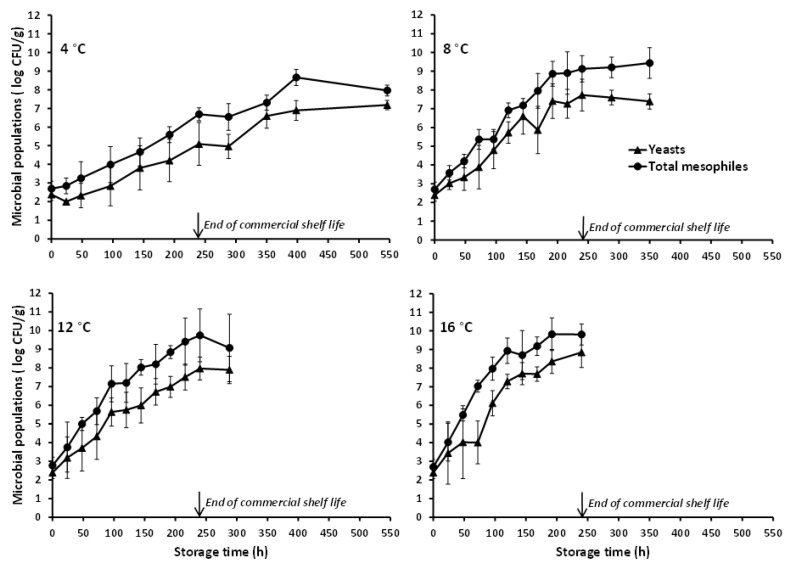
Mean (±standard deviation, *n* = 4) populations of yeasts and total mesophiles in uninoculated (control) battered chicken nuggets, stored at different isothermal conditions.

**Figure 3 foods-10-00533-f003:**
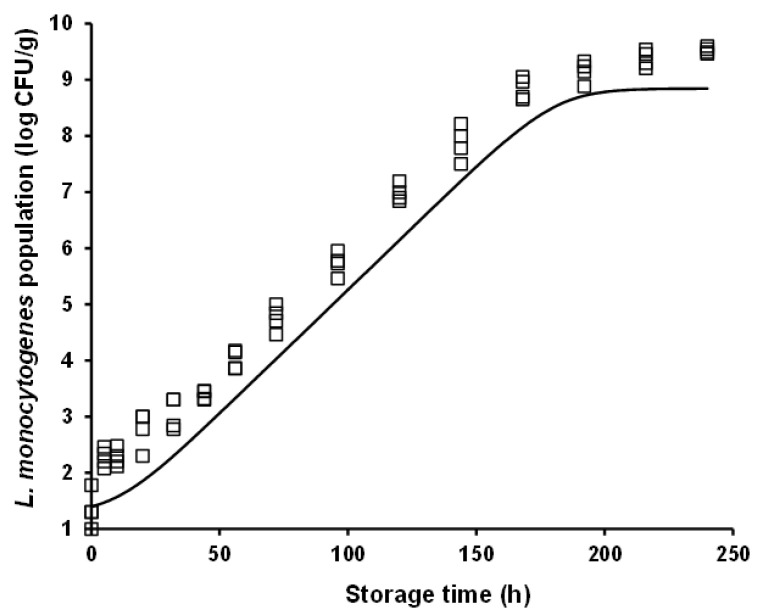
Observed (symbols) and predicted (line) growth of *Listeria monocytogenes* in battered chicken nuggets during storage at 10 °C.

**Figure 4 foods-10-00533-f004:**
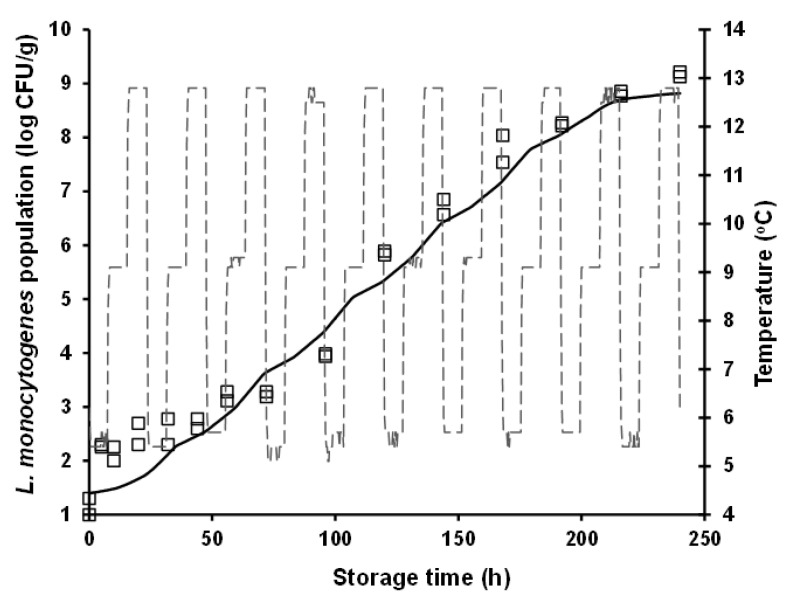
Observed (symbols) and predicted (continuous line) growth of *Listeria monocytogenes* in battered chicken nuggets during storage under dynamic temperature (dashed line) storage conditions (i.e., 8 h at 4 °C, 8 h at 8 °C, and 8 h at 12 °C).

**Table 1 foods-10-00533-t001:** Growth kinetic parameters of *Listeria monocytogenes* in artificially inoculated battered chicken nuggets stored at different temperatures.

Temperature (°C)	Growth Kinetic Parameter ^1^			Adjusted *R*^2^ (min–max) ^2^
	*λ* (h)	*μ_max_* (h^−1^)	*y_end_* (log CFU/g)	
4	21.80 ± 5.59	0.055 ± 0.005	− ^3^	0.962–0.985
8	17.56 ± 1.26	0.065 ± 0.020	8.93 ± 0.55	0.974–0.992
12	– ^3^	0.103 ± 0.130	8.95 ± 0.56	0.981–0.992
16	– ^3^	0.183 ± 0.055	8.65 ± 0.82	0.979–0.990

^1^ Values are means ± standard deviations (*n* = 4); *λ*: lag time; *μ_max_*: maximum specific growth rate; *y_end_*: maximum population density. ^2^ Range of the adjusted coefficient of determination (*R*^2^) values for the different product samples. ^3^ Not estimated by primary modelling.

## Data Availability

The data presented in this study are available on request from the corresponding authors. The data are not publicly available due to privacy restrictions.

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
