# Peer review of "Growth of Listeria monocytogenes in Partially Cooked Battered Chicken Nuggets as a Function of Storage Temperature"

_foods, 2021, doi:10.3390/foods10030533_

Round 1
Reviewer 1 Report
General Comments
The topic is interesting, as there is not so much in literature in relation to Listeria monocytogenes contamination of processed meat from poultry. It is actually possible that chicken nuggets are contaminated by Lm, because this pathogen is very frequent in raw poultry meat. Therefore it is interesting to understand the kinetics of this pathogen in processed poultry meat during storing, considering that it can grow at refrigeration temperature and, according to the initial contamination, could also get to high concentrations that could not be eliminated by a mild cooking process.
I see this study as a first step that should be followed by other studies on thermal inactivation of Listeria monocytogenes I this kind of product. This should be highlighted in the discussion.
In general, the manuscript is well written, methods are scientifically sound, but some details about methods used for the challenge test should be added. Moreover I suggest to improve introduction/discussion and add some more recent references on Listeria monocytogenes prevalence in raw chicken meat and poultry products, and on the persistence of this pathogen in the processing plants.
Specific comments
Line 54: not only live birds are an important reservoir of Lm: this pathogen has been recently found in raw chicken meat from Lm-negative flocks, highlighting the importance of persistence in the slaughterhouse environment (e.g. “Listeria monocytogenes in poultry: Detection and strain characterization along an integrated production chain” Food Microbiol 2020)
Line 55: these are mostly studies on AMR, it could be possible to speculate also to other recent studies on the high prevalence of Lm in raw chicken meat and meat products, and on the persistence of this pathogen in chicken meat slaughterhouses and processing environments (e.g. “Significance and Characteristics of Listeria monocytogenes in Poultry Products” Int J Food Sci 2019)
Line 61-64: it could be useful to note that a Listeriosis outbreak has been recently linked to cooked chicken meat (e.g. “Listeria monocytogenes in Cooked Chicken: Detection of an Outbreak in the United Kingdom” J Food Prot 2020).
Lines 76-85: Usually mixtures with less strains are used (2 or 3). Please explain why you used a 6 strains mixture (e.g. to mimic the natural multistrain contamination that could be found in a processing environment)
Lines 133-135: in materials and methods it is clear that there were two batches treated as two separate experiments. But results are presented without distinction, as they were obtained from a single batch/experiment. Please clarify.
Lines 230-233: it is not clear how the results differed between the two batches. Are the means referred to all the samples? The conditions between the two batches/experiments could have changed, therefore the results should be presented separately. In alternative, you should provide data to prove that there were no significant differences between the two batches.
Lines 446-447: please speculate also on possible further studies to evaluate the effect of undercooking on Lm in chicken nuggets (temperature, duration of cooking etc.)
Author Response
The topic is interesting, as there is not so much in literature in relation to Listeria monocytogenes contamination of processed meat from poultry. It is actually possible that chicken nuggets are contaminated by Lm, because this pathogen is very frequent in raw poultry meat. Therefore it is interesting to understand the kinetics of this pathogen in processed poultry meat during storing, considering that it can grow at refrigeration temperature and, according to the initial contamination, could also get to high concentrations that could not be eliminated by a mild cooking process.
I see this study as a first step that should be followed by other studies on thermal inactivation of Listeria monocytogenes I this kind of product. This should be highlighted in the discussion.
The authors thank the reviewer for his/her comments. The need for further studies on the thermal inactivation of L. monocytogenes has been acknowledged in the discussion of the revised manuscript, in the last paragraph of section 3 (p. 11), through the addition of the following part: “Nevertheless, in order for reliable exposure assessment information to be compiled, the collected data should be evaluated in tandem with data on the thermal inactivation behavior of the pathogen in the studied product. Based on the growth data collected herein, future research should evaluate the thermal inactivation kinetics of L. monocytogenes in battered chicken nuggets, taking into account different scenarios in terms of initial contamination levels and cooking regimes (i.e. time-temperature combinations corresponding both to cooking instructions and undercooking).”
In general, the manuscript is well written, methods are scientifically sound, but some details about methods used for the challenge test should be added. Moreover I suggest to improve introduction/discussion and add some more recent references on Listeria monocytogenes prevalence in raw chicken meat and poultry products, and on the persistence of this pathogen in the processing plants.
The reviewer’s comment was taken into account and additional more recent references on the pathogen’s prevalence in raw poultry products, as well as on its persistence in processing plants have been added in the revised manuscript. Please see references #17 (Iannetti et al., 2020) and 18 (Jamshidi and Zeinali, 2019) in p.13 of the revised manuscript.
Specific comments
Line 54: not only live birds are an important reservoir of Lm: this pathogen has been recently found in raw chicken meat from Lm-negative flocks, highlighting the importance of persistence in the slaughterhouse environment (e.g. “Listeria monocytogenes in poultry: Detection and strain characterization along an integrated production chain” Food Microbiol 2020)
The reviewer’s comment was taken into account and the suggested reference has been included in the “Introduction” of the revised manuscript (reference #17), and the corresponding sentence has been updated to the following: “Beyond live birds, which constitute an important reservoir of L. monocytogenes [15], reports on high prevalence of the pathogen in raw chicken products, along with concerns of increasing antibiotic resistance [4,16–19], demonstrate that its association with poultry meat production should not be regarded as trivial from a public health perspective.” (p.2 of the revised manuscript).
Line 55: these are mostly studies on AMR, it could be possible to speculate also to other recent studies on the high prevalence of Lm in raw chicken meat and meat products, and on the persistence of this pathogen in chicken meat slaughterhouses and processing environments (e.g. “Significance and Characteristics of Listeria monocytogenes in Poultry Products” Int J Food Sci 2019)
The reviewer’s comment was taken into account and the suggested reference has been included in the “Introduction” of the revised manuscript (reference #18), and the corresponding sentence has been modified to the following: “Indeed, L. monocytogenes may establish niches in poultry slaughterhouses and processing environments, serving as persistent sources of cross-contamination of end products, especially under conditions of poor compartmentalization of the processing line [18,20,21].” (p.2 of the revised manuscript).
Line 61-64: it could be useful to note that a Listeriosis outbreak has been recently linked to cooked chicken meat (e.g. “Listeria monocytogenes in Cooked Chicken: Detection of an Outbreak in the United Kingdom” J Food Prot 2020).
The comment was taken into account, and the aforementioned reference on the L. monocytogenes outbreak linked to the consumption of cooked chicken has been included in the revised manuscript as reference #27 (p. 13). Please see the part “Yet, as suggested by epidemiological investigations, cooked poultry products may also be involved in listeriosis outbreaks [27]…” in p.2 of the revised manuscript.
Lines 76-85: Usually mixtures with less strains are used (2 or 3). Please explain why you used a 6 strains mixture (e.g. to mimic the natural multistrain contamination that could be found in a processing environment)
According to guidelines and recommendations particularly with regard to food safety challenge testing for L. monocytogenes, multiple-strain composites of 3-5 strains should be used for inoculation purposes in order to account for variation in the growth/survival behavior among strains of the pathogen (unless the behavior of individual strains is known and, in such case, selection of strain(s) with robust growth or inactivation characteristics under harsh environmental conditions is advisable). This was the reason for using the six-strain mixture in this study, and a corresponding brief explanation has been included in the revised manuscript as “The abovementioned strains were used as a multiple-strain composite in this study in order to account for variation in growth among L. monocytogenes strains, in line with pertinent recommendations and guidelines with regard to food safety challenge studies [42–44].”, while references #42, 43 and 44 have been added (p. 14).
Lines 133-135: in materials and methods it is clear that there were two batches treated as two separate experiments. But results are presented without distinction, as they were obtained from a single batch/experiment. Please clarify.
The two distinct products batches and the correspondingly different inoculation/storage experiments were used as biological replicates, whereas the duplicate samples analyzed within each experiment at each sampling internal were used as technical replicates, resulting in a sample size of n=4 (product inoculation for model calibration purposes as well as for monitoring the evolution of the natural microbiota in uninoculated product). This is articulated in the yellow-highlighted part in p. 4, while some additions for clarification purposes have also been added in the revised manuscript; please see the part “Battered chicken nuggets corresponding to two distinct product batches were artificially inoculated and stored as described above, in the course of two independent experimental replicates conducted at different time periods, allowing for biological variation to be taken into account.” in the last paragraph of section 2.2 (p. 3), as well as the “n=4” addition in the first paragraph of section 3.1 (p. 5). This information is also available in the captions of Figures 1 and 2.
Lines 230-233: it is not clear how the results differed between the two batches. Are the means referred to all the samples? The conditions between the two batches/experiments could have changed, therefore the results should be presented separately. In alternative, you should provide data to prove that there were no significant differences between the two batches.
Please see the authors’ response to the previous comment and the provided clarifications. The idea was exactly to take into account in the modelling procedure the biological variability encountered between different product batches and experimental procedures (such as the one involved in the preparation of cultures for inoculation purposes). The experimental conditions (protocols, procedures, equipment and applied storage temperatures) were the same when treating the two product batches, and the collected data were provided in conjunction in order for the normally anticipated biological variability to be taken into account.
Lines 446-447: please speculate also on possible further studies to evaluate the effect of undercooking on Lm in chicken nuggets (temperature, duration of cooking etc.)
The reviewer’s comment was taken into account. Please see the authors’ response to the first comment.
Reviewer 2 Report
The research here proposed, attempt to predict the behaviour of Listeria monocytogenes in fresh, partially cooked breaded chicken nuggets conventionally refrigerated, as well as in condition of thermal abuse. Samples were artificially inoculated at low level with a mixture of six Listeria monocytogenes strains. The first aspect to point out is that authors cannot be sure that all the six strains really shared the same behaviour, since no tracking of the inoculum was carried out… The graphical trends are actually the sum of the growth curves of six different microbial cultures and this undeniably jeopardize the model they provide.
In addition, authors monitored the dynamics of the main microbial groups likely occurring on poultry meat, but not the main one: psychrophilic bacteria. Honestly I find hard to understand why TSA agar plates were incubated at 30°C. Listeria may grow on PCA and, in such conditions the pathogen may reasonably grow faster than bacteria usually developing on refrigerated meat products. In this regard I think it is mandatory to add yeast loads in figure 1 graphs. In light of the above, the scenario depicted 257-261 puzzles me…
At any rate, the work is well written but needs to be re-organized: some details are pleonastic, several concepts are more than once repeated, and too many references are used to remark concepts that are widely known.
With reference to the manuscript organization, I warmly suggest to merge the sections Results and Discussion, to avoid reiterations. Actually, the sentence at lines 359-356 is a repetition of concepts already reported in the Introduction, while the succeeding section (Lines 357-368) would be better placed in the Introduction. The concept at lines 441-447 is a repetition as well.
Lines 198-202: This part of the experimental plan should be detailed in the preceding section.
Delete figure 3. Data can be reported in the text.
Examples of unnecessary details: lines 92-95; lines 117-119; lines 120-121; lines 142-143; lines 156-160.
Examples of redundant references: two references for stating that Listeria monocytogenes is a foodborne pathogen of public health significance, two for saying that it is ubiquitous and psychrotrophic, three for saying that it is associated to poultry products. This is probably too much…
On the other hand, I would give a look of the following review article:
Khalid, Tahreem, et al. Review of Quantitative Microbial Risk Assessment in Poultry Meat: The Central Position of Consumer Behavior. Foods 9.11 (2020): 1661.
Minor remarks
Line 96: Is it ‘composites’ the best term?
Line 102: Probably the acronym CFU does not need explanations any longer.
Lines 167-170: Better placed in the Discussion section.
Line 264: According to routine procedure , Salmonella spp. should be reported as absent. If further tests have been performed to search S. enterica they need to be reported in Material and Methods.
Please add ‘data not shown’ at line 265.
Line 338. Such sentence needs to be related to the strains used in the trials.
Lines 339-340. Listeria loads are around 1 log CFU/g in figure 1.
Author Response
The research here proposed, attempt to predict the behaviour of Listeria monocytogenes in fresh, partially cooked breaded chicken nuggets conventionally refrigerated, as well as in condition of thermal abuse. Samples were artificially inoculated at low level with a mixture of six Listeria monocytogenes strains. The first aspect to point out is that authors cannot be sure that all the six strains really shared the same behaviour, since no tracking of the inoculum was carried out… The graphical trends are actually the sum of the growth curves of six different microbial cultures and this undeniably jeopardize the model they provide.
The authors agree with the reviewer. Indeed, using strain mixtures in food safety challenge studies aiming at model development inevitably results in speculative predictions regarding the behavior of foodborne pathogens, unless molecular approaches (e.g., pulsed-field gel electrophoresis) allowing for strain differentiation and monitoring are simultaneously applied. This has been observed and discussed in a previous study carried out in our laboratory (cited as reference #45 in the revised manuscript). However, the concomitant conductance of molecular analyses was not feasible in the context of the present study. Furthermore, if single strains have not been studied individually so that single-strain selection is based on robust growth/survival characteristics allowing for the worst-case scenario assessment, it has been recommended that multiple-strain composites are used so that variation among L. monocytogenes strains is taken into account (please see references #42, 43 and 44 in p. 14 of the revised manuscript). In any case, and in order for this limitation to be acknowledged, the part “Moreover, it should be acknowledged that although allowing for the integration of potential growth variation among L. monocytogenes strains, the multiple-strain composite utilized in the present study results in model predictions that cannot be linked to specific strain(s) of the organism. Molecular monitoring of the evolution of L. monocytogenes strains comprising the applied inoculum in food safety challenge studies has demonstrated that the pathogen’s strains dominating during storage may be considerably affected by the environmental conditions (e.g., temperature), highlighting the importance of strain selection in quantitative microbiology studies [45].” has been included in the revised manuscript (p. 11).
In addition, authors monitored the dynamics of the main microbial groups likely occurring on poultry meat, but not the main one: psychrophilic bacteria. Honestly I find hard to understand why TSA agar plates were incubated at 30°C. Listeria may grow on PCA and, in such conditions the pathogen may reasonably grow faster than bacteria usually developing on refrigerated meat products. In this regard I think it is mandatory to add yeast loads in figure 1 graphs. In light of the above, the scenario depicted 257-261 puzzles me…
The authors agree with the reviewer in that the important group (for a chilled product) of “psychrophilic bacteria” were not enumerated in this work. Nonetheless, the long incubation periods (10-15 days) of plates required for the reliable enumeration of this group, was beyond the scope of this study, whose main objective was the quantitative description of the growth of L. monocytogenes. Nonetheless, since L. monocytogenes is a psychotropic bacterium, it is highly likely that it would also grow during such long incubation periods, with the abovementioned issue of interfering with the psychrophilic bacteria also being probable. At any rate, a pertinent comment has been made in the revised manuscript; please see the part “Since psychrophilic bacterial populations were not enumerated in the framework of the present work, future studies, specifically designed and conducted for this purpose, should contribute to the in-depth characterization of the microbial association of this product and its evolution during storage.” (p.7).
With regard to adding yeast counts in Figure 1, the authors do not think that it would be useful for the following two reasons:
- Due to the very similar microbial counts attained on TSA and RBC agar plates (as stated in the yellow-highlighted part in p. 6-7 in the revised manuscript), the corresponding curves would overlap.
- In the monocytogenes inoculation experimental trials, the samples were only occasionally (i.e. at five selected time intervals during storage) analyzed on RBC agar, with the complete storage period being covered in the case of the uninoculated product (results presented in Figure 2). In order for this information to be present, a pertinent addition has been made in the revised manuscript [p. 4: “…(iii) Rose Bengal Chloramphenicol (RBC) agar (Lab M Limited) for the determination of the populations of molds and yeasts after incubation of plates at 25 °C for 5 days (at selected time intervals during storage).”]
At any rate, the work is well written but needs to be re-organized: some details are pleonastic, several concepts are more than once repeated, and too many references are used to remark concepts that are widely known.
The reviewer’s comment was taken into account and the recommended modifications have been made in the revised manuscript. Please see the authors’ responses to the specific comments outlined below.
With reference to the manuscript organization, I warmly suggest to merge the sections Results and Discussion, to avoid reiterations. Actually, the sentence at lines 359-356 is a repetition of concepts already reported in the Introduction, while the succeeding section (Lines 357-368) would be better placed in the Introduction. The concept at lines 441-447 is a repetition as well.
All suggested modifications have been made in the revised manuscript. The results and discussion have been merged, the above mentioned section has been moved to the “Introduction” (see the paragraph “Although the survival, growth and control…has not been previously assessed” in p. 2), while repetitions have been removed from the discussion of the manuscript.
Lines 198-202: This part of the experimental plan should be detailed in the preceding section.
The authors do not agree with this suggestion, since this part refers to the additional experimental trials conducted specifically for the purpose of model validation (i.e. collection of independent data that would be used to externally validate the model under both constant and dynamic temperature conditions). In this sense, and in line with similar descriptions in multiple predictive modelling studies in the scientific literature, this experimental information has been maintained under section 2.4 of “Materials and Methods” (Modelling of L. monocytogenes growth).
Delete figure 3. Data can be reported in the text.
Taking into account the reviewer’s recommendation, Figure 3 of the original submission has been deleted in the revised manuscript, and the pertinent information is reported solely in the text of the manuscript (see yellow-highlighted part in section 3.2, p.8 of the revised manuscript).
Examples of unnecessary details: lines 92-95; lines 117-119; lines 120-121; lines 142-143; lines 156-160.
All the aforementioned parts have been removed in the revised manuscript, excluding the last one (lines 156-160 of the original manuscript, now seen as a green-highlighted section in p. 4 of the revised manuscript) which, according to the authors’ view, includes information (not previously mentioned in the text) which is important and should be kept, since it describes the microbiological analyses carried out in the uninoculated (with L. monocytogenes) chicken nuggets.
Examples of redundant references: two references for stating that Listeria monocytogenes is a foodborne pathogen of public health significance, two for saying that it is ubiquitous and psychrotrophic, three for saying that it is associated to poultry products. This is probably too much…
Taking into account the reviewer’s comment, the references for stating the well-established ubiquitous and psychrotrophic character of L. monocytogenes have been removed and only one reference was kept to state the public health significance of the organism (the one referring to the EU). Nonetheless, in order to also take into account the comments of Reviewer #1 with regard to sufficiently reporting the incidence of the organism in various poultry products, these references were kept (while some additional have been also included in the revised manuscript, again as part of Reviewer #1 recommendations; yet these additions were tried to be kept to the minimum).
On the other hand, I would give a look of the following review article: Khalid, Tahreem, et al. Review of Quantitative Microbial Risk Assessment in Poultry Meat: The Central Position of Consumer Behavior. Foods 9.11 (2020): 1661.
The authors thank the reviewer for this recommendation. This very interesting recent publication (initially missing the authors’ attention) has been included in the revised manuscript and cited as needed (see pp. 2, 6).
Minor remarks
Line 96: Is it ‘composites’ the best term?
“Composites” has been changed to “mixtures” in the revised manuscript (see p. 3).
Line 102: Probably the acronym CFU does not need explanations any longer.
The explanation of the acronym “CFU” has been removed in the revised manuscript.
Lines 167-170: Better placed in the Discussion section.
Taking into account the reviewer’s comment, this part has been moved to the “Results and Discussion” section of the revised manuscript (see p. 7).
Line 264: According to routine procedure , Salmonella spp. should be reported as absent. If further tests have been performed to search S. enterica they need to be reported in Material and Methods.
Although there are actually only two species under the genus Salmonella, namely Salmonella enterica and S. bongori, and only the former is known for its foodborne transmission, the reviewer’s comment was taken into account and “S. enterica” was changed to “Salmonella spp.” In the revised manuscript (see p. 7)
Please add ‘data not shown’ at line 265.
The phrase “data not shown” has been added in the revised manuscript (p. 7).
Line 338. Such sentence needs to be related to the strains used in the trials.
The reviewer’s comment was taken into account, and the corresponding sentence has been modified to the following: “The microbiological data collected under the conditions of this study (and for the tested L. monocytogenes strains) demonstrate that, even in the case of relatively low-level contamination and under well-controlled refrigeration (simulated by the isothermal stor-age at 4 °C herein), L. monocytogenes may proliferate in battered chicken nuggets in the course of their shelf life to levels potentially hazardous for susceptible population groups, particularly in the case of highly virulent strains of the organism” (p. 6, below Figure 1 in the revised manuscript).
Lines 339-340. Listeria loads are around 1 log CFU/g in figure 1.
Τhe authors cannot comprehend the connection between the abovementioned line numbers (as provided in the pdf of the manuscript sent by the Editorial office) and the raised comment. Assuming that this is a typo, and that what the reviewer actually wanted to comment on was the 1.40 log CFU/g mentioned in section 3.2, we would like to mention the following: this value refers to the mean initial population as estimated from all the conducted experimental trials (including the additional experiments conducted for model validation purposes) and not only the initial levels determined in the experiments conducted for model calibration and which are depicted in Figure 1 as ca. 1 log CFU/g. In order for this to be clarified, the corresponding sentence has been revised to “In both cases, as initial microbial population was used the mean value of the actual initial inoculum concentrations, determined by plate counting across all inoculation experimental trials including the independent model validation trials (i.e. 1.40 log CFU/g)…” (see yellow-highlighted part in p. 9 of the revised manuscript).
Reviewer 3 Report
The authors have described an assessment of the growth of a cocktail of Listeria monocytogenes strains inoculated on to the surface of partially cooked chicken nuggets. The referee has some methodological queries and presentational and feels the presentation would be much improved with some additional data relating this to retail/catering practices, and comparison of this paper to existing models. Specific comments are listed below.
- Title: it would be helpful to readers if the title could reflect a bit more accurately the product. The products are partially cooked and appears to be battered not breaded (see later note). It would also be helpful to indicate that these products investigated are not intended as ready-to-eat.
- Abstract, line 18). It would also be helpful to state here the temperatures of storage that were modelled.
- Some discussion about the normal practices for storage would be helpful here. These products are commonly retailed as frozen and certainly breaded frozen chicken have been associated with salmonellosis outbreaks in Canada, USA and Europe. Although frozen products have been associated with listeriosis (e.g. meatballs and corn), this is less well understood than refrigerated products: frozen products do not provide opportunities for L.monocytogenes to grow during storage. There are ready-to-eat breaded chicken products on sale in Europe and these are often retailed at refrigerated storage, it is the referee’s experience that the ready-to-cook products are more often sold frozen. Since L.monocytogens was not detected in the uninoculated products tested here (L.monocytogenes is rather common in raw chicken), the major risks appear to be from cross contamination. With the frozen products associated with outbreaks, data from USA, Canada and Europe showed the presence of Salmonella (unlike the products described here): presumably L. monocytogenes would survive the manufacturing process too. If these frozen products were defrosted, there is a possibility of growth both from L.monocytogenes surviving the manufacturing process as well as from cross contamination on the surface (as modelled here). Hence the risks may be underestimated. Some discussion about the normal storage conditions at retail (how were the products packaged and were modified atmospheres used), together with an assessment of the appropriateness of the 10 day shelf life mentioned in the discussion (can it be stated if this is from the day of production?).
- Introduction line 52. Outbreaks of human listeriosis associated with consumption of chicken products have been reported (and recently reviewed), and not just sporadic cases as stated in the text.
- Materials and Methods, line 1-2. The authors state that the inoculum was designed to give 1-2 log CFU/g of the chicken products. On Fig 1 this looks consistently like 1 log (10 organisms). Can the authors comment on this? Does this mean there is some die off of the organisms. There are no error bars at time zero and minimal up to 50 hours at the two lower temperatures. Can the authors comment on this, particularly around stochastic behaviour at this level (see later point).
- Materials and methods line 108. The product is described as a ‘breaded’ chicken breast. Bread does not appear in the list of ingredients (lines 110-113), and since the authors describe batter (line 111 with flour and eggs), it would seem more appropriate to describe this as a battered and not a breaded product.
- Materials and Methods line 171. Were salt concentrations or aw measured for this product? This might allow comparison of the outputs of the growth models here with other modelling approaches and establish if there are major differences.
- Table 1. To aid comprehension without consulting the body-text, could the various parameters (ÊŽ, μmax, R2 etc) be explained as a footnote for this Table. Could an explanation as to why no growth parameters were calculated for the 12â—¦C and 16â—¦C temperatures.
- Figs 1and 2. The x axes are different and vary from 500, 300 and 250 hours. To allow meaningful comparison, could these be drawn to the same scale or at least an explanation as to why these are different? Since the shelf life for this product is 10 days, could this also be marked on this figure.
- Fig 1. Could the authors comment on a lack of a lag phase before growth of monocytogenes occurs at all temperatures except 4â—¦C.
- Fig 5. A maximum growth of about 109 monocytogenes was detected under the dynamic temperature experiment which varied between about 5.5â—¦C and 13â—¦C. This is about the same level as at 12â—¦C isothermal experiments. Can the authors comment that the lower refrigeration temperatures appear to have little effect on when the maximum concentration is reached.
- Line 399. The inoculation level is measured at about 10 cfu/g. The authors rightly point out that populations may behave stochastically at <100cells. Was there any evidence of stochastic behaviour in these experiments?
Author Response
The authors have described an assessment of the growth of a cocktail of Listeria monocytogenes strains inoculated on to the surface of partially cooked chicken nuggets. The referee has some methodological queries and presentational and feels the presentation would be much improved with some additional data relating this to retail/catering practices, and comparison of this paper to existing models. Specific comments are listed below.
The authors thank the reviewer for his/her comments. Indeed, comparison with the few existing models (although referring to distinct in terms of technology and/or storage conditions products compared to the product studied herein) has been made (please see yellow-highlighted part in p. 11 of the revised manuscript). With regard to more detailed responses to specific comments of the reviewer, these are provided below.
Title: it would be helpful to readers if the title could reflect a bit more accurately the product. The products are partially cooked and appears to be battered not breaded (see later note). It would also be helpful to indicate that these products investigated are not intended as ready-to-eat.
Taking into consideration the reviewer’s comment, the title of the revised manuscript has been changed to “Growth of Listeria monocytogenes in partially cooked battered chicken nuggets as a function of storage temperature”.
Abstract, line 18). It would also be helpful to state here the temperatures of storage that were modelled.
The applied storage temperatures, taken into account in model development (i.e. 4, 8, 12 and 16 °C), have been provided in the abstract of the revised manuscript.
Some discussion about the normal practices for storage would be helpful here. These products are commonly retailed as frozen and certainly breaded frozen chicken have been associated with salmonellosis outbreaks in Canada, USA and Europe. Although frozen products have been associated with listeriosis (e.g. meatballs and corn), this is less well understood than refrigerated products: frozen products do not provide opportunities for L.monocytogenes to grow during storage. There are ready-to-eat breaded chicken products on sale in Europe and these are often retailed at refrigerated storage, it is the referee’s experience that the ready-to-cook products are more often sold frozen. Since L.monocytogens was not detected in the uninoculated products tested here (L.monocytogenes is rather common in raw chicken), the major risks appear to be from cross contamination. With the frozen products associated with outbreaks, data from USA, Canada and Europe showed the presence of Salmonella (unlike the products described here): presumably L. monocytogenes would survive the manufacturing process too. If these frozen products were defrosted, there is a possibility of growth both from L.monocytogenes surviving the manufacturing process as well as from cross contamination on the surface (as modelled here). Hence the risks may be underestimated. Some discussion about the normal storage conditions at retail (how were the products packaged and were modified atmospheres used), together with an assessment of the appropriateness of the 10 day shelf life mentioned in the discussion (can it be stated if this is from the day of production?).
The authors agree with the above points raised by the reviewer. Certainly, the fate of L. monocytogenes (a well-known psychrotrophic organism) in battered/breaded chicken nugget products, although not studied as much as other foodborne pathogens, is anticipated to greatly depend on the type of product (frozen or fresh and chilled) as well as on its expected distribution in the food supply chain. It is true that partially cooked (non-ready-to-eat) battered/breaded poultry products (including chicken nuggets) distributed under refrigeration are not as commonly marketed in the EU and the United States as frozen ready-to-cook or fresh (chilled) RTE products. In order for this to be acknowledged, a pertinent commentary has been included in the revised manuscript via the following part in the last paragraph of the “Results and Discussion” section: “With the majority of the battered/breaded chicken nugget products marketed in the EU and the United States being distributed either as ready-to-cook frozen products or as RTE refrigerated products, the product category investigated herein (i.e. partially cooked but non-RTE product distributed under refrigeration) has not been amply represented in food safety challenge studies.” (p. 11 of the revised manuscript). Regarding the commercial shelf life of the studied product, yes it refers to the period starting from the day of production, and this can be accurately claimed here since the product samples arrived at the laboratory and used in the experiments within 24 h from production. For this to be clarified, the phrase “from the day of production” has been added in the revised manuscript (see section 2.2 in p. 3), while (in line with a following comment of the reviewer) the end of the product’s shelf is indicated in Figures 1 and 2 to allow for a better evaluation of the collected data.
Introduction line 52. Outbreaks of human listeriosis associated with consumption of chicken products have been reported (and recently reviewed), and not just sporadic cases as stated in the text.
Taking into account the reviewer’s comment, the corresponding sentence has been revised to “Both sporadic and epidemic listeriosis have been associated with poultry production and processing environments as well as with various poultry products” along with the appropriate in-text citations (see the first sentence of the second paragraph in p. 2 of the revised manuscript).
Materials and Methods, line 1-2. The authors state that the inoculum was designed to give 1-2 log CFU/g of the chicken products. On Fig 1 this looks consistently like 1 log (10 organisms). Can the authors comment on this? Does this mean there is some die off of the organisms. There are no error bars at time zero and minimal up to 50 hours at the two lower temperatures. Can the authors comment on this, particularly around stochastic behaviour at this level (see later point).
The goal was to achieve an inoculation level slightly higher than 1 log CFU/g (ca. 1.5 log CFU/g) so that a real-life contamination scenario could be assessed to the greater possible extent, while at the same time avoiding (as much as possible) the technical difficulties and the uncertainty associated with low bacterial levels. Since differences within 1 log are not regarded as significant from a biological standpoint in microbiological studies, this inoculation goal was communicated as 1-2 log CFU/g. Nonetheless, it should be noted that although in the experimental trials conducted for model calibration the attained inoculation level was indeed ca. 1 log CFU/g (with very low standard deviation, if any, and that is why it cannot be seen in Figure 1), a wider range of inoculation levels were recorded in the trials conducted for model validation (ranging from ca. 1 to 1.7 log CFU/g). The mean value of all initial levels was 1.40 log CFU/g, as mentioned in the yellow-highlighted part in p. 9 of the revised manuscript. We cannot know if this range (which by no means is significant from a biological perspective) is indicative of some cells dying off, but we agree that it may demonstrate to some extent the stochastic behavior exhibited by bacteria at low cell concentrations.
Materials and methods line 108. The product is described as a ‘breaded’ chicken breast. Bread does not appear in the list of ingredients (lines 110-113), and since the authors describe batter (line 111 with flour and eggs), it would seem more appropriate to describe this as a battered and not a breaded product.
The authors thank the reviewer for this comment, which was taken into account and the word “breaded” has been replaced by “battered” when referring to the studied product throughout the manuscript.
Materials and Methods line 171. Were salt concentrations or aw measured for this product? This might allow comparison of the outputs of the growth models here with other modelling approaches and establish if there are major differences.
The aw of the product was not measured/monitored during storage in the context of the present work. The only information that we had at our disposal with regard to the salt content of this product was the one provided in the nutrition information on the commercial label of the product, namely 1% (w/w) sodium and 2.5% salt. Given this, as well as the fact that the study was not designed to include aw as a model parameter, comparison of the developed growth model with other models taking into account this parameter, cannot be made.
Table 1. To aid comprehension without consulting the body-text, could the various parameters (ÊŽ, μmax, R2 etc) be explained as a footnote for this Table. Could an explanation as to why no growth parameters were calculated for the 12â—¦C and 16â—¦C temperatures.
The reviewer’s comment was taken into account and the suggested information has been included as footnote in Table 1 in the revised manuscript. Also, regarding the 12 and 16°C, it is not true that no growth parameter were estimated, but only the lag time could not be determined via primary modelling (due to the fast growth of the pathogen at the higher storage temperatures). In order for this to be clear the corresponding footnote in Table 1 has been modified to “Not estimated by primary modelling”, while a comment regarding the absence of lag time at 12 and 16°C has also been included in the text of the manuscript (see p. 8 of the revised manuscript).
Figs 1and 2. The x axes are different and vary from 500, 300 and 250 hours. To allow meaningful comparison, could these be drawn to the same scale or at least an explanation as to why these are different? Since the shelf life for this product is 10 days, could this also be marked on this figure.
The reviewer’s comment was taken into account, and the recommended modifications in Figures 1 and 2 have been made in the revised manuscript.
Fig 1. Could the authors comment on a lack of a lag phase before growth of monocytogenes occurs at all temperatures except 4â—¦C.
Since L. monocytogenes is a psychrotrophic organism, which means that in some food products with favorable intrinsic parameters (e.g., neutral pH, low microbial antagonism), such as the one studied herein, grows relatively fast even at slightly abusive refrigerated temperatures (e.g., 8 °C), it not easy to observe graphically the lag time (which anyway is in the hour-scale and actually lower than 20 h). The provided in Table 1 lag times for 4 and 8°C are the output of primary modelling, which could not be provided (due to the rather fast growth of the organism) for the 12 and 16°C based on the sampling time intervals applied in this work (as also discussed in previous comment).
Fig 5. A maximum growth of about 109 monocytogenes was detected under the dynamic temperature experiment which varied between about 5.5â—¦C and 13â—¦C. This is about the same level as at 12â—¦C isothermal experiments. Can the authors comment that the lower refrigeration temperatures appear to have little effect on when the maximum concentration is reached.
Taking into account the above comment, a pertinent comment has been included in the “Results and Discussion” section of the revised manuscript. Specifically, the following part has been added: “It is interesting to note that the rather high L. monocytogenes population (ca. 9 log CFU/g) observed at 240 h of storage (approximating the end of the product’s commercial shelf life) at the abusive constant temperatures of 12 °C (Figure 1) and 10 °C (Figure 3), was also attained during storage for the same time period under dynamic temperature conditions (Figure 4). The latter observation is indicative of the detrimental impact that cold chain temperature fluctuations may have on the growth of foodborne pathogens and ultimately, on the safety of chilled food products.” (see p.9 of the revised manuscript).
Line 399. The inoculation level is measured at about 10 cfu/g. The authors rightly point out that populations may behave stochastically at <100cells. Was there any evidence of stochastic behaviour in these experiments?
To the authors view, a strong indication of the potentially stochastic behavior of the organism at the low inoculation levels applied herein, is the fact that rather frequently the recorded growth curves for the different samples (n=4) were considerably different (even between technical replicates corresponding to the same product batch), despite the fact that the initial inoculation level was very similar. This is also demonstrated by the error bars in the L. monocytogenes counts illustrated in Figure 1. Actually, the observed differences was the reason that the secondary model was not fitted to the growth rate data using all the (individual) μmax values estimated from primary modelling for each one of the tested samples, but alternatively, and in order for this to be overcome, the secondary modelling was performed based on the estimated mean μmax values at each storage temperature.
Round 2
Reviewer 2 Report
None